# Role-play simulations as an aid to achieve complex learning outcomes in hydrological science

Arvid Bring[1,2], Steve W. Lyon[1,3]

[1]Department of Physical Geography, Stockholm University, SE-106 91, Sweden
[2]Bolin Centre for Climate Research, Stockholm University, SE-106 91, Sweden
[3]The Nature Conservancy, Delmont, New Jersey, 08314, USA

*Correspondence to*: Arvid Bring (arvid.bring@natgeo.su.se)

**Abstract.** Students in hydrology are expected to become proficient in a set of quantitative skills, while also acquiring the
ability to apply their problem-solving abilities in real-life situations. To achieve both these types of learning outcomes, there
is broad evidence that activity-based learning is beneficial. In this paper, we argue that role-play simulations in particular are
useful to achieve complex learning outcomes, i.e., making students able to coordinate and integrate various analytical skills in
complicated settings. We evaluated the effects of an integrated water resources management (IWRM) negotiation simulation
next to more traditional teaching methods intended to foster quantitative understanding. Results showed that despite similar
student-reported achievement of both complex and quantitative intended learning outcomes, the students favored the
negotiation simulation over the traditional method. This implies that role-play simulations can motivate and actively engage a
classroom thereby creating a space for potential deeper learning and longer retention of knowledge. While our findings support
the utility of simulations to teach complex learning outcomes and indicate no shortcoming in achieving such outcomes next to
traditional methods aimed at quantitative learning outcomes, simulations are still not widely used to foster activity-based
learning in the classroom. We thus conclude by presenting three particularly challenging areas of role-play simulations as
learning tools that serve as potential barriers to their implementation and suggest ways to overcome such roadblocks.

## 1 Introduction

As the societal need and student demand for science education continues to require interdisciplinarity, higher-education
practitioners are increasingly faced with teaching and achieving complex sets of intended learning outcomes in the classroom.
Education in water resources and hydrology is an example of such a situation as water connects across several biophysical
aspects and societal problems (Di Baldassarre et al., 2013; Bring et al., 2017; Rodell et al., 2018; Strandmark et al., 2015).
Hydrology students today move on to careers that span a wide spectrum of skills: some become niche experts in quantitative
modeling, while others end up in policy work as generalists. All hydrology students, however, should acquire a set of core
quantitative skills that make them professionals in understanding the functioning of water in the earth system (Lyon and
Teutschbein, 2011).

At the same time, mastering quantitative skills is not enough. In order to solve, or even partly address, almost any science-based or environmental problem today, the quantitative skills must be complemented with an ability to navigate administrative, political, social and economic constraints. Environmental systems are complex, dynamic, and highly non-linear (Liu et al., 2007), and therefore, no single real-world solution exists outside a context. Besides conveying the quantitative skills, a fundamental component of learning in hydrology and environmental sciences is the appreciation of this context (McClain et al., 2012; Pahl-Wostl et al., 2007). In the end, students should become proficient not only in analytically solving a problem, but also defending their arguments, understanding those of others, and considering the importance of (and potential conflicts or trade-offs between) various political, economic and social aspects. We are increasingly asking our students to engage with and achieve complex and cross-disciplinary intended learning outcomings in our environmental classes (Lyon et al., 2013).

Today, the need for complex skills is strongly emphasized in the environmental science community (McClain et al., 2012; Seibert et al., 2011; Wagener et al., 2012). This emphasis reflects a societal development since the middle of the twentieth century, in which not only teaching and learning, but the role of environmental science professionals, has transformed thoroughly (Sivapalan, 2018). This applies in particular to a discipline such as hydrology, which has evolved from its ancient roots in applied engineering to a science that is today central in Earth system science, geosciences and environmental sciences. While the engineering perspective still remains strong in hydrology, compared with other Earth system science-related disciplines, making connections to the wider system context in hydrology teaching has clearly grown in importance. One aspect which reflects this is the limited use of textbooks to teach hydrology (Wagener et al., 2007), which may reflect that they contain the theory but not the interdisciplinary context within which many students will work. Today, we are not only increasingly viewing the environment as a complex system, which requires us to revise what we teach students about how the system works, we are also required by advances in the science of learning and behavior to revise how we view teaching, and what methods that best reinforce learning (DeNeve and Heppner, 1997).

## 1.1 Role-play simulations as a useful method to achieve complex intended learning outcomes

As the content of environmental science curricula have developed to include a broader context of problem-solving (e.g., Lyon and Teutschbein, 2011), simulations provide an opportunity to illustrate and test social and political consequences of environmental problems. Role-play simulations can be thought of as evolving and open-ended interactive problems where students have specific roles with responsibilities and constraints, and experience the effect of their actions and decisions (Gredler, 2004). They create an active environment in the classroom that has been linked to higher achievement and better non-cognitive outcomes (D'Angelo et al., 2014). In particular, role-play simulations provide a condensed way to introduce natural science students to a set of aspects often not previously included in their training, such as economic considerations, the importance of political context, participatory processes, and societal tradeoffs.

Role-play simulations are also used outside academia, as training tools for professional water managers and other stakeholders. In such a context, simulations can help facilitate a process of "social learning" (Pahl-Wostl et al., 2007), in which participants together move towards a shared vision and understanding of a water management problem, and in which their previously established views are challenged. For example, a government-employed water manager might act in the role-play simulation as a vocal "green" non-governmental activist, and gain understanding for such a stakeholder's perceptions and main interests. There is some evidence that stakeholder perceptions have shifted due to such simulations (Pahl-Wostl and Hare, 2004). As water resource management has moved from sectoral decision-making towards integrated water resources management (IWRM), both role-play simulations and other types of participatory and collaborative modelling approaches have become increasingly popular (Basco-Carrera et al., 2017; Lyon et al., 2018). Furthermore, participatory approaches in general are supported both formally by the Dublin Principles (http://www.wmo.int/pages/prog/hwrp/documents/english/icwedece.html) and implicitly by indications that active stakeholder participation in modelling increases likelihood of reaching consensus (Basco-Carrera et al., 2017; Loucks and Van Beek, 2017).

Today's learning paradigm emphasizes the importance of learners performing and practicing the tasks they are expected to learn when they have completed the course. In essence, the activities of students throughout a course should be constructively aligned with the intended learning outcomes (Biggs, 2003), so that students practice the abilities specified in the course's intended learning outcomes during the course. Also in this regard, simulations can allow for implicit advantages with regards to the current understanding of how learning takes place, as they allow participants the opportunity to exercise the complex skills (e.g., management, negotiations, arguing, evaluating options, considering values) they are intended to master as per the intended learning outcomes.

The literature on simulations in teaching is growing rapidly. A Web of Science search on the topic "role-play simulations" in higher education research during the ten-year period 2008-2017 returns more than 200 publications – an increase of over 500% over the preceding ten-year period where the count was less than 50 publications.

In environmental science, simulations as learning tools align with the system dynamics approach to conceptualizing and understanding the environment itself. In systems dynamics, complex environmental problems are modeled as sets of non-linear interactions between multiple processes and states, and allow exploration of system responses to various actions (Winz et al., 2009). The similarity between a modeling tool used to understand the field's object of study itself and role-play simulations as learning tools indicates that the latter may map well onto our understanding of the environment as a complex system, and therefore be appropriate tools in education about the system.

In an examination of a suite of learning simulations in environmental education, all dealing with turn-based resource allocation problems, Bazan (1976) noted a bimodal response distribution, where one group of students eventually came to accept selfish

(but environmentally detrimental) behavior by participants, on the grounds that everyone was acting that way. Students in the other group of response types were more troubled when confronted with the dysfunction of efforts to manage the environmental system, and were concerned about what to do about it. This illustrates that simulations are effective in illuminating the complexity of environmental systems, opening students' eyes to unexpected outcomes, and reinforcing students' abilities at

navigating them. However, Bazan (1976) notes that formal evaluation of the effectiveness of the teaching method was lacking.

Perhaps less encouraging is an early experiment that evaluated simulations against conventional instruction in schools, which showed no effect on measured learning (Fennessey et al., 1974). In fact, their review of outcomes at the time indicated that lectures had been more effective than simulations in teaching concepts and attitudes; however, this may reflect a lack of

complex intended learning outcomes. To this point, Schnurr et al. (2013) detailed experiences from teaching an environmental negotiation simulation on biological diversity for undergraduate students. Their goals with the simulation included teaching students both the "complex and interrelated dynamics" of international environmental agreements, as well as providing them with professional skills "such as mediation, lobbying, consensus building, public speaking, as well as environmental policy writing and evaluation". Their own experience of the effectiveness of the simulation was that it was good for producing learning

outcomes separate from the traditional quantitative outcomes, and they report at 71% agreement from students in course evaluations that the simulation was effective for learning.

Further, the experience of Rusca et al. (2012) indicated that simulation games were highly useful in fostering a "T-shaped professional", that is, someone with a deep knowledge in their specific field but with good abilities to work efficiently in cross-

disciplinary teams. At the same time, Rusca et al. (2012) note a number of limitations with simulations, not the least the demands on the teacher. Managing a role-play simulation requires strong focus of the teacher on knowledge construction by students themselves, and the teacher must him- or herself be an experienced T-shaped professional in order to guide students to appropriate material. This perceived extra effort has likely been a hindrance to the implementation of simulations largely across the sciences and when faced with cross-disciplinary outcomes where they may serve better than traditional siloed

lecturing approaches.

From this perspective, in this study we evaluate an attempt to use a negotiation simulation exercise to achieve complex learning outcomes in an environmental science course. Specifically, we consider an advanced hydrology course at the Department of Physical Geography at Stockholm University. In our experiment, we evaluate the success of using an integrated water resources

management (IWRM) negotiation simulation while also at the same time using a suite of more traditional teaching exercises that focus on the water balance and aim to provide students with quantitative skills. Our main aim is to determine how simulations have been perceived by students as a learning method, compared with traditional teaching methods, and how students rate their achievement of both traditional, quantitative learning objectives as well as more integrative and complex learning objectives.

## 2 Methods

Our evaluation of the negotiation simulation builds on its inclusion into the introductory course in a Master's level program in hydrology, hydrogeology and water resources (https://www.natgeo.su.se/english/education/courses-programmes/master-programmes/hydrology-hydrogeology-and-water-resources/master-s-programme-in-hydrology-hydrogeology-and-water-resources-1.53631). We draw upon 6 independent offerings of the course over the years 2011 to 2017 to provide data in this study summarizing across some 93 students. Throughout these various offerings, the main professor responsible for the course has been the same person (Prof. Steve W. Lyon, Stockholm University). The role-play simulation was taught by Arvid Bring (experienced teacher in negotiations) in 4 years out of 6, while Johan Kuylenstierna (experienced teacher in negotiations) and Steve Lyon (novice teacher in negotiations) taught one offering each. The structuring and instruction while not a constant has by and large been consistent. As such, while a protentional source of error, we do not expect that the variability between offerings had a significant impact on our data.

In the following we describe the role-play simulation, which was designed to deepen students' understanding of the complex technical, social, managerial, and environmental aspects of managing international water resources. We then contrast this simulation exercise with a more traditional approach to teaching hydrology, consisting of a set of projects focusing on quantitatively estimating the water balance and hydrological flows in a catchment. Lastly, we outline the metrics used to assess student-rated achievement of the intended learning outcomes for the course and their perceptions of the teaching methods considered.

### 2.1 IWRM negotiation simulation

The IWRM negotiation simulation "Transboundary waters – resolving conflicts and building trust" was developed by Johan Kuylenstierna as a tool to train water managers and professionals at the Stockholm International Water Institute (SIWI). Since then, it has also been extensively used in higher education, and is a recurring component of the Master's course *Local to Global Water Vulnerability and Resilience* at the Department of Physical Geography at Stockholm University, Sweden.

The simulation is structured as a ten-actor negotiation role-play, with representatives from various government ministries, local authorities, and non-government organizations. The actors together form a temporary river basin commission between two fictitious countries sharing a watercourse. Water quantity is not a central issue in this basin, but water quality is a major concern due to pollution from industries and agriculture. The goal of the game is to negotiate an agreement on how the permanent river basin commission should be organized, in order to efficiently solve the main water problems in the basin. Key decision items for establishing the suggested commission include defining the territorial scope of the commission, its guiding principles, membership and governance, immediate key actions, financing, and enforcement responsibilities. Decisions on quantitative water allocations or water quality measures are thus not elements of the game, but qualitative discussion about

such issues form the basis for negotiations on how to manage the water. After initial preparations, students carry out pre-negotiations during a full day, with one-on-one meetings under the overarching supervision of one of the students, who is assigned the head of delegation of the host country and therefore chairs the meeting. The second day is devoted to six hours of formal negotiations, during which the chair and participants deliberate over the key decision areas outlined above and detailed in the student instructions. After a break during which the chair and potential rapporteur prepare to present the outcome, in the form of a text of the negotiated agreement, a two-hour debriefing session ends the simulation. The debriefing starts with a recapitulation of the outcome of the meeting, after which students form small groups and discuss their experiences for about ten minutes, with ensuing discussions of main points in plenary.

## 2.2 Traditional teaching approach – water balance projects

An important aim of the course is the ability to evaluate the available water resources for an area and quantitatively estimate a water balance. The principal teaching method to this end, next to a number of introductory lectures, is a set of two projects where students perform catchment-scale water resource assessments.

The first project is an individual assignment and starts with students identifying a basin of their own choice. Students need to characterize the region in terms of climatic setting and delineate the geographical extent. Students are then required to find adequate data for quantifying precipitation, runoff, and potential and actual evapotranspiration over the basin. They need to compute the long-term average water balance, evaluate whether there are any changes in storage, and if so, provide an estimate of those. The second project also departs from establishing a water balance, but this time as a group exercise. Students work in teams to quantify additional hydrological processes, such as agricultural, industrial, and domestic water flows. Although the projects have been slightly adjusted during 2011-2017, the structure has been the same. The objectives have always been to establish the water balance, quantify changes in storage or climate, and estimate numbers for additional hydrological flows impacting the basin.

## 2.3 Comparatively assessing teaching methods side by side

To evaluate the success of the negotiation simulation, we analyzed student surveys from six course occasions during 2011-2017 (in 2015 no course was given). The surveys have been designed to both probe student satisfaction with various components of the course, and to evaluate how well the students perceive that they have reached the intended learning outcomes (ILOs).

As our analysis builds on self-reported achievements, we cannot consider the responses a metric of how well students perform against some external standard, but we nevertheless consider their own responses a relevant indicator of how well they achieve the goals. We opt for this approach as there is no possibility (due to a double-blind student evaluation approach) to directly connect performance to individual perceptions. Although this may be a limitation in terms of determining their achievement

objectively, no such limitation exists in measuring their satisfaction with the teaching methods, as we sample their own impressions directly, albeit anonymously. Ideally, we would design an experiment where one group of students is exposed to the "treatment" of the role-play simulation, while the other group only receives traditional instruction and can thus act as "control". Unfortunately, such an experiment would be impractical for the teacher and require large groups of students in order to minimize the impact of other variability in and between student groups. However, we will now start sampling "before" and "after" perceptions, to test if the role-play simulation changes student perceptions of water management negotiations. Although that question is not in focus in this current study, and we have no data to evaluate yet, we could possibly extract more information on the impact of the negotiation simulation compared to traditional teaching methods in future versions of the survey.

The form and exact wording of the evaluation questions contained in the survey has varied slightly over the studied period. For this study, we consider two direct questions on how well students approved of a) the IWRM negotiation simulation and b) the quantitative water balance projects. We also consider two other questions relating to the students' own appreciation of how well they achieved the c) complex ILO and d) quantitative ILO of the course. In all, these four questions (Table 1) allow us to test both how well students liked the respective teaching methods, and how well they think they have achieved the corresponding learning outcomes.

In Table 1, we summarize the questions we have investigated, and a concise shorthand version we will use in the remainder of this paper. Students have provided answers on a five-graded scale from 1 to 5, with 1 equivalent to "Strongly disagree" and 5 equivalent to "Strongly agree".

In addition to the specific questions, some years also had an open-ended question on what students thought was the best part of the course. Although these responses can be considered anecdotal and less representative, they can still reveal impressions of students and teachers that convey relevant information, and we therefore summarize these comments below.

## 3 Results

### 3.1 Quantitative responses

Figure 1 shows the distribution of the pooled student responses ($N$ responses = 91 +/- 1 for all questions) across the six course occasions (patterns for individual years are largely similar). In summary, the role-play simulation is as effective at teaching complex learning outcomes as the traditional teaching methods are at teaching quantitative learning outcomes. The responses to questions about the project work ("Approved of quantitative project work", orange bars in Fig 1) show a peak at 4, indicating agreement, with the second-most common response being 5, indicating strong agreement. For responses concerning the

appreciation for the simulation exercise, a rating of 5, indicating strong agreement, was the clearly predominant response, with 4 being the second-most common response.

Overall, for project work, 78% of respondents were unambiguously positive (rating 4 or 5), while the corresponding figure for
the simulation exercise was 87%. The largest difference between the distribution of responses were for the rating of 5, with 60% of respondents giving the simulation that rating, compared to 34% for the project work. The mean of the two distributions is also significantly different (Mann Whitney U two-tailed test, $p < 0.01$), with the simulation rated 4.4 on average, compared with an average of 4.0 for the projects.

In Figure 2, we show levels of student-rated achievement of the intended learning outcomes (ILOs). Overall, students generally perceive that they have reached the ILOs. Out of the 91 +/- 1 answers for each of the ILOs, there were only two responses of 2 or below, indicating disagreement. In contrast, 94% of respondents agreed that they had achieved the Quantification ILO, and 87% agreed that they had achieved the Complex ILO (responses of 4 or 5). Differences between the Quantification ILO (orange bars in Figure 2) and the Complex ILO (blue bars) are relatively small and not significant (Mann-Whitney U two-
tailed test, $p > 0.05$).

Compared to the approval ratings distribution, the main difference is a shift from 5 to 4 for the self-rated achievement of the Complex ILO, so that the distribution more closely resembles the distribution for the Quantification ILO. There is also a shift in the lower end of the scale (3 and below), at which level more students rate their achievement of the Complex ILO compared
to the Quantification ILO.

## 3.2 Teacher and student reflections

In the teachers' experience, the negotiation simulation has worked well and has become a well-integrated and appreciated part of the course. A strength with the role-play simulation is that it allows teaching of a very broad set of topics related to hydrology and water management in a brief span of time (the in-class time for the simulation is roughly two 8-hour days). Another
strength is that students attempt to solve what they perceive as a real-world problem. Most students tend to take the game very seriously, and work hard to act in their role and to achieve a solution for the group. The game also tends to awaken interest in the complex reality in solving water resources problems, particularly for students with natural science backgrounds, who sometimes tend to overestimate the power of rational thought and argument in solving politically sensitive questions. This is useful from the perspective of the entire Master's program. Specifically, since the negotiation simulation game is connected
to the first course in the Master's program, this critical thinking and interest in complex problems shines through (and increases engagement) in subsequent courses and final thesis projects.

A weakness encountered from the teachers' perspective and echoed by the students, is that the game is not based on a real-world example, which sometimes causes confusion as to whether facts and arguments about the water resources situation in the targeted basin are realistic. For example, a student group could "make up" artificial hydrological or climatological data to support their claims and another student is forced to respond. Simply put, such a situation typically breaks the "fourth wall" of the simulation and require teacher intervention, which we typical seek to avoid in the real-time of the negotiation simulations. Another weakness is that the various roles to be played may give students somewhat different opportunities to learn. For instance, it is more "in character" for some roles to argue and voice strong or one-dimensional opinions, whereas others may have less influence or clear perspectives. While this is by design based on the variations in stakeholder power and influence associated with water resource negotiation, it can bring about differentiation in the overall experience for the students. Also, we have noted that some students find the experience taxing on social interactions or classroom dynamics thus creating an uncomfortable environment. To address this, we have over the years shifted the simulation to the later part of the course to facilitate students in any cohort to get an opportunity to know each other better before participating in a role play.

Nevertheless, students mostly give positive comments about the game in open-ended questions about the course. Specifically, the most common answer to the open-ended question "What did you like best about this course" (not posed all years, $N = 30$) was "the negotiation simulation" (or equivalent). When details are provided, students particularly highlight that the simulation experience provided them with an opportunity to gain experience with a real-world application of problems faced in water resource management and an understanding for how difficult it is to achieve consensus across conflicting interests when not only physical aspects of hydrology are included.

Of course, although most students describe the simulation in positive terms, a few have voiced negative reflections. These comments were typically related either to the experience itself, which a few students found too challenging, or to the utility of the game for achieving what they wanted to learn from the course. One student in particular expressed discontent with having to learn about fictitious countries, since there would be no benefit of having studied the material after the simulation was over.

## 4 Discussion: Challenges and Lessons Learned

Although the teacher and location of the negotiation simulation has changed over the six course occasions, the simulation was ranked better than the traditional learning activity, in terms of average student approval, in four years out of six. Even in years when the simulation was supervised by a less experienced teacher, student approval of the simulation, as well as student achievement of the Complex ILO, was not lower than in years taught by a more experienced teacher (Mann Whitney U one-tailed test, $p > 0.05$). This is a positive around the ability of novice or inexperienced educators to incorporate simulations in the classroom – there is a gentle learning curve with the students.

Furthermore, our results indicate that there is no inherent conflict between a traditional and more guided teaching method, used to teach quantitative skills, and the independent, activity-based learning that a simulation provides. This points to the possible benefits of including learning situations such as simulations and games, as this would complement and not interfere with students reaching the quantitative ILOs. However, a caveat is that our study only involved teaching the methods side-by-side. There may thus be a latent conflict in the possible alternative uses of the time spent in the negotiation simulation, which could have included, for instance, more in-depth training of quantitative skills. At the same time, the discourse on teaching in environmental sciences in general, including hydrology, has evolved in such a way that it is difficult to motivate an exclusive focus on quantitative skills in higher education. From that perspective, our results at least indicate that some introduction of dedicated teaching methods for complex learning outcomes does not interfere with quantitative skills achievement. In summary, experience from half a decade of simulation-based learning in hydrology at Stockholm University indicates that role-play simulations can be useful tools for students to achieve complex learning outcomes and broaden their understanding of the importance of the context for solving environmental issues.

Although our results indicate that simulations are highly appreciated as learning tools, they are naturally not adequate in all situations, and even when useful, there remain several challenges with using them in teaching. From personal experience, we have found three top challenges to be particularly salient. In the next sections, we provide examples of these top challenges – and suggest some possible ways to address them as lessons learned.

### 4.1 Acquiring theoretical knowledge through simulations

One potential problem with simulations is their adequacy for achieving theoretical knowledge. Typically, simulations are thought of as a way to master the application and use of knowledge, but not necessarily as a way to also learn facts and understand their context. There is a distinction in which the simulation process is understood as consisting of two separate components: (1) acquiring the theoretical framework outside of the simulation and through other teaching methods, and (2) applying the theoretical knowledge and learning how to use it in the simulation. Although not always explicitly acknowledged, this perceived dichotomy is evident in many descriptions of simulations in the literature (DeNeve and Heppner, 1997; Rusca et al., 2012; Schnurr et al., 2013).

Although simulations are often perceived – foremost at least – as ways to test and apply previously acquired knowledge, we believe the incorporation of new theoretical knowledge can also be achieved with appropriate design. For instance, it may be possible to design the simulation so that acquisition of facts and theory are essential to success. This could be achieved with instructions that participants will be evaluated (possibly by their peers to facilitate active learning, e.g., Lyon and Teutschbein (2011)) on factual understanding and use of theoretical knowledge. Although such instructions may interfere with other objectives, such as a full focus on the student's role in the simulation process, they may still be useful if the trade-off is deemed acceptable.

Another way with which factual and theoretical learning could be encouraged is more in line with the simulation itself. Depending on the designated roles, participants could be encouraged to act as experts in their field, and that they, in order to fulfil that role, must prepare extensively by studying hard on their own. Such an approach has worked well in the climate change negotiation exercise "World Climate Simulation" (www.climateinteractive.org/programs/world-climate/). Participants could also be encouraged to seek facts to support their arguments, or seek out videos, fact sheets, or other brief and accessible material that they actually use in their role to convince other students in the simulation.

Yet another way uses our knowledge of learning in general to design incentives. Students could be rewarded for preparation reading with quizzes, bonus exam points, or other ways to encourage their deep learning in preparations. Although this approach does not use the simulation moment itself as the mode of learning, it could be designed to integrate well with the simulation. Depending on the context, the incentives could themselves be presented as parts of the simulation.

## 4.2 Quality learning in student groups with mixed disciplinary backgrounds

In all groups, there is some variation in the background of students, but in some courses, this variation can be quite large, and include students from both natural and social sciences (which is the case for the course considered in this research).

In the particular context of an environmental negotiations simulation, there are many dimensions of knowledge that the experience-based learning approach can deliver (DeNeve and Heppner, 1997; McClain et al., 2012; Pathirana et al., 2012). Although it is not certain that the activity will lead to learning across all of these broader dimensions, there is a possibility that students acquire a deeper understanding of natural science facts, environmental processes, and engineering principles, as well as social science facts and theories, including economic trade-offs, political rules and procedures, and negotiation skills. In addition, generic skills such as meeting planning, time management, people management, proper professional conduct, and the like can also be acquired.

How can learning such a diverse set of knowledge and skills be effective for groups of students with various disciplinary backgrounds? One possible way may be to carefully consider the educational background of students when assigning roles and responsibilities. This can be considered on two levels: content of instructions (focus on subject matters), and on mandate and formal authority. A student that should ideally learn more about politics may be placed in a role where they have to carefully study these aspects to play their role adequately. In contrast, a student with experience in political science, but less in engineering or natural science, may be assigned a role as environmental non-government representative or technical expert. Although this may place students outside their comfort zone, it ensures that the path of least resistance is not taken, and that students will have to engage more deeply in their preparations in order to succeed. It also provides an opportunity for students to familiarize themselves with knowledge, arguments, and perspectives of professions other than the one they themselves are

likely to end up in. This may be particularly useful as they will likely have to engage with colleagues from other professions throughout their career.

Another approach may be to allow students to choose roles themselves, deciding what aspect they would most prefer to spend
effort in developing, with the aid of descriptions of what characteristics are most important for the different roles. A risk with this option is that most students will indeed take the path of least resistance and choose according to what they already do well. However, depending on their motivation, the approach may be justifiable if the student will then have a conscious intention to enhance and deepen their skill in a particular field.

### 4.3 Formative assessment in simulations

Assessment provides the fundamental feedback to the student about their learning. Assessment can be categorized into two types: formative assessment, which serves to provide feedback *during* learning and acts as an indicator and guide to the student to help them with monitoring their learning, and summative assessment, which provides an evaluation *after* learning of how well the student has performed in relation to set standards (see Biggs, 2003).

Although formative assessment has an important role in traditional teaching methods, its role in simulations is less clear. The simulation may be conceived as an act of formative assessment in itself, but this simplifies the problem. How can student learning be monitored and reinforced throughout the simulation, providing adequate feedback to the students?

Debriefing is one way of formative assessment. The debriefing is both convergent, in that it evaluates student performance
against some predetermined standards or objectives, and divergent, in that it encourages students to reflect on their experiences and their learning (Rudolph et al., 2008).

However, debriefing is not the only type of formative assessment in a simulation. Evidence indicates that activation of students' metacognitive skills and reflection are an important aspect of learning (Nicol and Macfarlane-Dick, 2006). Therefore,
depending on the specific cases, there may be opportunities to enhance such metacognitive reflections, and at the same time provide formative feedback, during the course of a simulation.

A possible way is to introduce one or several "time-outs" during the course of the simulation. Although too frequent or extensive time-out may interfere with students' engagement in their roles, a reasonable amount of time-out could help students
regain some focus on their overarching goals with participating in the exercise, in contrast to their *in-simulation* goals.

For example, during a two-day simulation, the first day might end with a brief time-out session in which students work in pairs. Temporarily stepping out of their roles, students could be asked to identify the most important factors in the simulation

outcome thus far, and consider whether these factors will remain the same or change during the ensuing day. Furthermore, students may be asked to critically reflect on what they have done well during the first day, and how they intend to improve during the second day, in terms of their own learning and progress towards learning objectives. Again, the focus should not be on the in-simulation goals, but on their goals for their own learning.

Yet another way to include formative assessment in simulations is by designing the simulation so that progress depends on actions during the game, with evaluations along the way. In this manner, students would experience assessment in the form of repeated failures and successes, and hopefully improve in successive iterations. This approach, which aligns with Kolb's theory of experiential learning cycle stages, has been applied in a recently developed water safety planning game (Ferrero et al., 2018). However, not all types of complex skills lend themselves well to such a repeating structure within a single role-play simulation event.

## 5 Conclusions

In teaching hydrology, as is true across many environmental science disciplines, it is not enough that students master quantitative skills, as today's work in environmental science requires a number of more complex abilities, including an understanding of negotiations, policy, and economics. Activity-based learning tools such as role-play simulations can aid in achieving complex learning outcomes, and can be a useful complement to more traditional teaching methods designed to build proficiency in calculations and quantitative estimates. Despite a long history of this use in environmental science, and a suite of mostly positive accounts of effectiveness, there seems to be relatively little formal evaluation of simulations next to possible alternative learning approaches.

20

In this paper, we show that teaching a small negotiation exercise in a course on water resources helps students achieve complex learning outcomes, while not infringing on students' achievement of more traditional and focused learning outcomes. We also identify a suite of three key challenges to effective teaching using simulations, and outline a number of concrete suggestions as to how these challenges can be overcome. In summary, there are clear indications that role-play simulations can help in teaching the complex skills that today's students in environmental science will need, irrespective of the role they eventually will fill in their future careers.

25

## Data availability

To maintain the usefulness of the role-play described in this paper (confidentiality of instructions, actors and key elements of the game), it has not been published online, but details of the simulation and the investigation of student responses are available upon request to the authors.

30

## Competing interests

The authors declare that they have no conflict of interest.

## Author contribution

Conceptualization: AB and SWL; Funding acquisition: SWL; Methodology: AB and SWL; Formal analysis: AB; Investigation: AB and SWL; Writing: AB and SWL.

## Acknowledgements

We thank the European Union Erasmus+ (project 586345-2017) WATERMAS Water Management and Climate Change in the Focus of International Master Programs for partial support in completion of this research. The European Commission support for the production of this publication does not constitute an endorsement of the contents which reflects the views only of the authors, and the Commission cannot be held responsible for any use which may be made of the information contained therein.

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

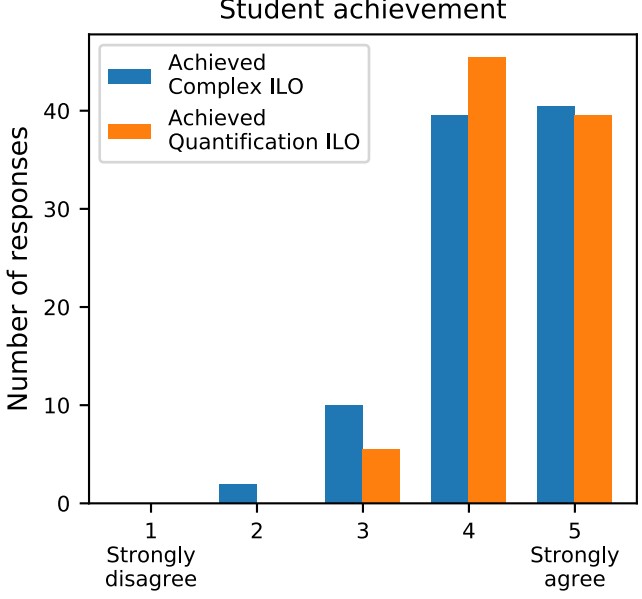

**Figure 1: Student approval ratings.** Distribution of pooled responses to student surveys over 2011-2017.

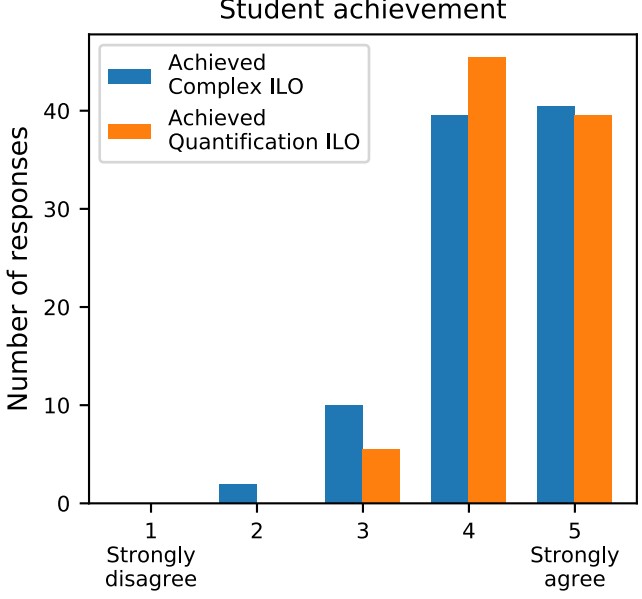

**Figure 2: Student self-reported learning achievement.** Distribution of pooled responses to student surveys over 2011-2017.

5    **Table 1. Questions on student surveys.**

| Questions on student appreciation of teaching activities | Short version |
| --- | --- |
| | |

| a) | The integrated water management negotiation game was useful and informative. | Approved of IWRM negotiation simulation. |
|---|---|---|
| b) | The projects on the water balance were useful and informative. | Approved of quantitative project work. |
| **Questions on student appreciation of reaching ILOs** | | **Short version** |
| c) | In this course, I learned better how to relate physical-chemical-geographical characteristics between upstream and downstream water to national and international water resource policy, environmental policy and cooperation-conflict management. | Achieved Complex ILO. |
| d) | In this course, I learned how to identify, extract and combine relevant information for analysis of water availability and quality across scales (local to global) with specific focus on constructing and constraining a water balance. | Achieved Quantitative ILO. |