# Peer review of "Role-play simulations as an aid to achieve complex learning outcomes in hydrological science"

_Hydrology and Earth System Sciences, 2018_

## Referee Comment (RC1) · Anonymous Referee #1 · 29 Jan 2019

This is an interesting paper on an educational topic that is a good fit for HESS. I have a few comments below that I think will deepen the argument of the paper and hopefully strengthen its case.

[1] Introduction: Paragraph two discusses the problem of placing hydrology into the right context within a classroom setting. I think that this is a very interesting issue, which could be discussed a bit further. Wagener et al. (2007) surveyed the hydrology teaching community and found that the use of hydrology textbooks (which contain theory, but little about how to place it into context) for classroom teaching is rather limited (as fraction of the material taught). A large reason for this is that we (as educators)

[Figure]

have to put a lot of effort into providing examples of context. The authors do this well in terms of the water management framing, but maybe they could expand this section a bit more for a broader discussion?

See: Wagener et al. (2007). Taking the pulse of hydrology education. Hydrological Processes, 21, 1789-1792.

[2] Section 1.1: I would find it useful to see more depth on the discussion of the role of participatory modelling in IWRM. I remember using such role-play games like the once discussed in this paper over 20 years ago in classes I took from Eelco van Beek. I am not trying to say that the content of this manuscript is not interesting or new, but rather that there exists a very rich literature of the use of such software not just to teach students, but also to teach stakeholders and others about how IWRM management works. I think the authors should go more into depth of reviewing how such tools are used more widely to broaden the appeal of their paper. There will also be a lot of literature on how successful such software is in changing the thinking of stakeholders, which might be useful to include in the discussion section of this educational paper.

See example references: https://www.sciencedirect.com/science/article/pii/S1364815217300890 https://unesdoc.unesco.org/ark:/48223/pf0000143430

[3] Results: What I am missing in this part is a before-after survey of the students. While I understand that the authors only have a single student group (so there is no benchmarking group). However, the students are at the Master's level and I assume it is not their first exposure to hydrology. Could the authors not use a survey including questions on how students see the role of stakeholder negotiations etc. and compare their perception before and after the course? Or at least it would be interesting to hear the students state how their perceptions might have changed from the work done in the course.

---

## Referee Comment (RC2) · Anonymous Referee #2 · 20 Feb 2019

General comments:  c The paper contributes to the evidence that simulation and role play teaching techniques provide additional value to graduate education, especially in an interdisciplinary topic like water management.  c This is valuable because as the authors note, "formal evaluation of the effectiveness of the teaching method is lacking".  c The description of the educational "experiment" and its outcomes is a little unclear and could use some editing.

Specific comments:

Introduction:  c Pg 3 Starting Line 15: Add some description of the purpose of the simulations to prepare the reader to comprehend the different student responses. It

could be something like, "In an examination of a suite of learning simulations where students made resource allocation decisions, ..." or something more appropriate to those simulations. • Pg 3 Line 24: consider replacing "knowledge" with "outcomes" • Pg 3 ine 26: everything after "outcomes" in the sentence is stated awkwardly and doesn't seem to add value. Consider removing. • Pg 3 Line 31: say something like "the simulation was good for producing learning outcomes separate from the traditional quantitative outcomes". Otherwise it seems like you could mean that this study showed that simulations also helped with quantitative skills. From reading the rest of the paper, I don't think you meant that. • Pg 4 Line 6: Great point about the need for T-shaped professors. • Pg 4 Starting line 10: Somewhere in the paragraph before the Methods or in the beginning of Methods state explicitly that you taught both the simulation experience AND the water balance projects in all classes throughout the years. A really robust experiment would be to see the outcome differences of doing the simulation OR something else. I assumed that was the case without the explicit description. Methods: • Pg 4 Line 22: Mention the other less experienced teacher that taught the class that you mention later in the paper.

IWRM negotiation simulation: • Pg 5 starting Line 1: Provide a full reference for the IWRM negotiation simulation. I wasn't able to find it on the internet with the information provided. • Pg 5 starting Line 6: Share a little bit about the topical content of the activity, like what they were specifically negotiating? Water allocations? Water quality regulations? • Pg 6 line 6: You mention later that you could implement formal quizzes to provide motivation for students to learn additional material for the activity. What do you think about using quizzes to test what students learned from the simulation activity? You could ask them questions about what complex challenges arise in the real world to see if they learned anything. Results: • Pg 6 Line 28: Insert summary of the findings. Basically, the simulation activity is as useful for teaching additional outcomes as the traditional teaching methods are useful for teaching quantitative outcomes. • It's interesting that the students would prefer for the activity to have direct application to the real world. That's something to keep in mind for designing the most

effective simulation activities. Discussion: • Pg 8 line 29: Can you speak to any conflict from the fact that you had to delete some content from a traditional course to make time for the negotiation simulation? What quantitative material was not covered that would have been in a traditional course? Again, a robust experiment would be to test the learning outcomes from teaching that material instead of the simulation and seeing if students had better quantitative outcomes. This is the problem some people have with training some students to be T-shaped. They may not be able to go as deep in their field. • Pg 9 Line 27: This has worked well in the activity designed to represent climate change negotiations called "World Climate Simulation". • Pg 10 Line 7-13: The challenge of trying to convey this learning outcome should be better stated. I think it would be something like: there is a potential for learning about other disciplines, but the activity does not guarantee it. • Formative assessment in simulations: If the learning outcomes align well with the simulation outcomes, experiencing failure or success incrementally in the activity could provide formative assessment.

Technical corrections: • For citations like the one on pg 3 line 27, take the authors names out of the parenthesis and only parenthesize the date since you are directly referring the name in the sentence. • Pg 4 line 22: I think you omitted "professor" between "main" and "responsible". • Pg 10 Line 4: The first sentence is redundant and could be removed.
* * *

---

## Author Comment (AC1) · 2 Apr 2019

We thank the reviewers for many comments that have helped us improve the paper. In the following, we reiterate the reviewers' comments, and directly below each comment present our response and explain how we have changed our manuscript (a version with track changes activated is available in the supplement).

— Anonymous Referee #1

This is an interesting paper on an educational topic that is a good fit for HESS. I have a few comments below that I think will deepen the argument of the paper and hopefully

strengthen its case.

[1] Introduction: Paragraph two discusses the problem of placing hydrology into the right context within a classroom setting. I think that this is a very interesting issue, which could be discussed a bit further. Wagener et al. (2007) surveyed the hydrology teaching community and found that the use of hydrology textbooks (which contain theory, but little about how to place it into context) for classroom teaching is rather limited (as fraction of the material taught). A large reason for this is that we (as educators) have to put a lot of effort into providing examples of context. The authors do this well in terms of the water management framing, but maybe they could expand this section a bit more for a broader discussion?

See: Wagener et al. (2007). Taking the pulse of hydrology education. Hydrological Processes, 21, 1789-1792.

RESPONSE: We have now expanded the discussion to also cover briefly the use of hydrology textbooks and putting teaching into context (P2, lines 15-20).

— [2] Section 1.1: I would find it useful to see more depth on the discussion of the role of participatory modelling in IWRM. I remember using such role-play games like the once discussed in this paper over 20 years ago in classes I took from Eelco van Beek. I am not trying to say that the content of this manuscript is not interesting or new, but rather that there exists a very rich literature of the use of such software not just to teach students, but also to teach stakeholders and others about how IWRM management works. I think the authors should go more into depth of reviewing how such tools are used more widely to broaden the appeal of their paper. There will also be a lot of literature on how successful such software is in changing the thinking of stakeholders, which might be useful to include in the discussion section of this educational paper. See example references: https://www.sciencedirect.com/science/article/pii/S1364815217300890 https://unesdoc.unesco.org/ark:/48223/pf0000143430

RESPONSE: This is an interesting point and we have now added a section where we discuss the use of simulations in IWRM outside university teaching (P3, lines 1-12).

— [3] Results: What I am missing in this part is a before-after survey of the students. While I understand that the authors only have a single student group (so there is no benchmarking group). However, the students are at the Master's level and I assume it is not their first exposure to hydrology. Could the authors not use a survey including questions on how students see the role of stakeholder negotiations etc. and compare their perception before and after the course? Or at least it would be interesting to hear the students state how their perceptions might have changed from the work done in the course.

RESPONSE: We acknowledge that a more powerful (and resource demanding!) experiment could involve a "control vs treatment" setup, and ideally then also randomize the student allocation to each group. We agree also that testing student perceptions before and after the role-play negotiation could reveal the impact of the exercise on those perceptions, and although that question is not in focus in this study, we will add it in the future. We have now added some text that acknowledges these points (P6-7 lines 32-6).

— Anonymous Referee #2

General comments: The paper contributes to the evidence that simulation and role play teaching techniques provide additional value to graduate education, especially in an interdisciplinary topic like water management. This is valuable because as the authors note, "formal evaluation of the effectiveness of the teaching method is lacking". The description of the educational "experiment" and its outcomes is a little unclear and could use some editing.

RESPONSE: We have now added text to clarify the setup of the study (P4 line 29), compared it with alternatives (P6-7 lines 32-6) and also clarified our presentation of the outcome (P7 lines 25-26).

— Specific comments:

Introduction:

Pg 3 Starting Line 15: Add some description of the purpose of the simulations to prepare the reader to comprehend the different student responses. It could be something like, "In an examination of a suite of learning simulations where students made resource allocation decisions, : : :" or something more appropriate to those simulations.

RESPONSE: We have now added a brief description of the nature of these simulations (P3 lines 32-33).

— Pg 3 Line 24: consider replacing "knowledge" with "outcomes"

RESPONSE: Done.

— Pg 3 Line 26: everything after "outcomes" in the sentence is stated awkwardly and doesn't seem to add value. Consider removing.

RESPONSE: Removed as suggested.

— Pg 3 Line 31: say something like "the simulation was good for producing learning outcomes separate from the traditional quantitative outcomes". Otherwise it seems like you could mean that this study showed that simulations also helped with quantitative skills. From reading the rest of the paper, I don't think you meant that.

RESPONSE: Agreed, we added text as suggested to clarify (P4 lines 13-14).

— Pg 4 Line 6: Great point about the need for Tshaped professors.

Pg 4 Starting line 10: Somewhere in the paragraph before the Methods or in the beginning of Methods state explicitly that you taught both the simulation experience AND the water balance projects in all classes throughout the years. A really robust experiment would be to see the outcome differences of doing the simulation OR something else. I assumed that was the case without the explicit description.

RESPONSE: Acknowledged that such a setup would have been ideal (see also response to Reviewer 1's comment above). We have now clarified that both the simulation and the traditional methods were given simultaneously on P4 line 29.

— Methods:

Pg 4 Line 22: Mention the other less experienced teacher that taught the class that you mention later in the paper.

RESPONSE: We have now clarified teacher roles on P5 lines 7-9.

— IWRM negotiation simulation:

Pg 5 starting Line 1: Provide a full reference for the IWRM negotiation simulation. I wasn't able to find it on the internet with the information provided.

RESPONSE: This role-play simulation has never been published online. Part of the reason is to maintain usefulness of the role-play (confidentiality of instructions, actors and key elements of the game). We have now clarified that the simulation details are available on request (P5 lines 23-25).

— Pg 5 starting Line 6: Share a little bit about the topical content of the activity, like what they were specifically negotiating? Water allocations? Water quality regulations?

RESPONSE: We have now clarified the key issues of the negotiations on P5 lines 28-31, P6 lines 2-4.

— Pg 6 line 6: You mention later that you could implement formal quizzes to provide motivation for students to learn additional material for the activity. What do you think about using quizzes to test what students learned from the simulation activity? You could ask them questions about what complex challenges arise in the real world to see if they learned anything.

RESPONSE: As also noted above, we are now introducing questions to test before and after perceptions of the negotiation simulation, which will allow us to verify students'

degree of knowledge of negotiations.

— Results:

Pg 6 Line 28: Insert summary of the findings. Basically, the simulation activity is as useful for teaching additional outcomes as the traditional teaching methods are useful for teaching quantitative outcomes.

RESPONSE: Summary inserted on P7 lines 25-26.

— It's interesting that the students would prefer for the activity to have direct application to the real world. That's something to keep in mind for designing the most effective simulation activities.

Discussion:

Pg 8 line 29: Can you speak to any conflict from the fact that you had to delete some content from a traditional course to make time for the negotiation simulation? What quantitative material was not covered that would have been in a traditional course? Again, a robust experiment would be to test the learning outcomes from teaching that material instead of the simulation and seeing if students had better quantitative outcomes. This is the problem some people have with training some students to be T-shaped. They may not be able to go as deep in their field.

RESPONSE: We have now added a discussion about this potential trade-off on P9 line 32, P10 lines 1-5.

— Pg 9 Line 27: This has worked well in the activity designed to represent climate change negotiations called "World Climate Simulation".

RESPONSE: We now note this point on P10 line 33, P11 line 1.

— Pg 10 Line 7-13: The challenge of trying to convey this learning outcome should be better stated. I think it would be something like: there is a potential for learning about other disciplines, but the activity does not guarantee it.

RESPONSE: We have now rephrased this and acknowledge the challenge on P11 line 15.

— Formative assessment in simulations: If the learning outcomes align well with the simulation outcomes, experiencing failure or success incrementally in the activity could provide formative assessment.

RESPONSE: We now acknowledge this on lines P13 lines 4-8.

— Technical corrections:

For citations like the one on pg 3 line 27, take the authors names out of the parenthesis and only parenthesize the date since you are directly referring the name in the sentence.

RESPONSE: Done.

— Pg 4 line 22: I think you omitted "professor" between "main" and "responsible".

RESPONSE: Done.

— Pg 10 Line 4: The first sentence is redundant and could be removed.

RESPONSE: Done.

Please also note the supplement to this comment:
https://www.hydrol-earth-syst-sci-discuss.net/hess-2018-618/hess-2018-618-AC1-supplement.pdf

**Supplement:**

[revised manuscript text omitted]

---

## Author Response (AR1)

**Response to editor**

Dear Dr Viviroli,

Thank you for the comments on our revised manuscript. Below we list the requests for changes, and reply with references to how we have changed the paper. In the following pages we attach a copy of the manuscript with all changes indicated.

*Editor comments:*

*- In reply to a reviewer comment, you added information on the topical contents of the activity on P5 L28–31. This is most welcome, but it sounded like forming the temporary river basin commission mainly involves administrative tasks. Along the original reviewer comment, could you clarify whether issues like water allocations and water quality are also negotiated?*

> RESPONSE: We have now elaborated on the topic of the simulation, and clarified that water allocations and water quality issues are not addressed quantitatively in the game, but that qualitative discussions about such issues form a basis for how to organize the river basin commission (P5 L27-P6 L1).

*- Please carefully check reference style, e.g. P4 L10 (to "…point, Schnurr et al. (2013) detailed…"), P4 L17 (to "…experience of Rusca et al. (2012) indicated…"), P10 L26–27 (to "(possibly by their peers to facilitate active learning, e.g., Lyon and Teutschbein, 2011)"). You mentioned in your reply that this was fixed, but it appeared unchanged in the revised manuscript. (\*)*

> RESPONSE: We apologize for these omissions – they have now (really) been corrected (see changes on P4 L10, 18, 20; P10 L30-31), and all references checked.

*- The first author name for the reference on P1 L26 should be "Di Baldassarre".*

> RESPONSE: Corrected.

*- Please complete the following sections required by the journal: Data availability (along the lines of what you added to P5 L23–25, or directly moving that), Author contributions, Competing interests.*

> RESPONSE: We have now added these sections to the manuscript (P13 L27-P14 L5).

[revised manuscript text omitted]